## THE NATURAL HISTORY OF MODEL ORGANISMS

# Amphioxus as a model to study the evolution of development in chordates

**Abstract** Cephalochordates and tunicates represent the only two groups of invertebrate chordates, and extant cephalochordates – commonly known as amphioxus or lancelets – are considered the best proxy for the chordate ancestor, from which they split around 520 million years ago. Amphioxus has been an important organism in the fields of zoology and embryology since the 18th century, and the morphological and genomic simplicity of cephalochordates (compared to vertebrates) makes amphioxus an attractive model for studying chordate biology at the cellular and molecular levels. Here we describe the life cycle of amphioxus, and discuss the natural histories and habitats of the different species of amphioxus. We also describe their use as laboratory animal models, and discuss the techniques that have been developed to study different aspects of amphioxus.

**SALVATORE D'ANIELLO\*, STEPHANIE BERTRAND, HECTOR ESCRIVA\***

**\*For correspondence:**
salvatore.daniello@szn.it (SD'A);
hector.escriva@obs-banyuls.fr
(HE)

**Competing interest:** The authors declare that no competing interests exist.

## Introduction

Cephalochordates, commonly known as amphioxus or lancelets, belong to the monophyletic group of chordates, which also includes tunicates and vertebrates (*Figure 1A*). Amphioxus are marine benthic animals that feed on phyto- and zooplankton by filtering the seawater, and they are distributed worldwide in sandy habitats of tropical and temperate seas (*Bertrand and Escriva, 2011*).

A fascinating drawing by the Italian scientific illustrator Comingio Merculiano in the late 18th century captures the lifestyle of amphioxus (*Figure 2*). Unlike the adult, the embryos and larvae are planktonic and, depending on the species, the larval phase can last up to several months in the open sea. Therefore, amphioxus have a high potential for offshore larval dispersion in new coastal areas until they undergo the process of metamorphosis and become juveniles. The juveniles already show the typical adult body plan and, at this stage, they adopt a benthic lifestyle, prevalently buried in the substrate.

The name cephalochordate (i.e., cephalo- (head) and -chordate (notochord)), which was proposed by Ernst Haeckel in the 1860s (*Haeckel, 1866*), does a good job of describing the peculiarity of their anatomy, with the notochord extending to the front of the animal, beyond the cerebral vesicle (i.e. the most anterior structure of the central nervous system). The anatomy of cephalochordates is considered vertebrate-like, but simpler, having a prototypical chordate body plan. Chordate synapomorphies, present in amphioxus and vertebrates, include a dorsal hollow nerve chord and notochord, pharyngeal slits, segmented muscles and gonads, post anal tail, and homologs of pronephric kidney, pituitary and thyroid (*Figure 1B*). However, some typical vertebrate characteristics are not present in amphioxus such as paired sensory organs (image-forming eyes or ears), paired appendages and migrating neural crest cells. Their embryonic development includes 10 developmental periods, from the zygote to the adult (*Carvalho et al., 2021*; *Bertrand et al., 2021*), which are extremely well conserved among different amphioxus species.

An interesting anatomical feature of amphioxus development concerns their symmetry. In fact, larvae are completely asymmetrical, with the mouth and anus on the left side of the body. However, this asymmetry mostly disappears during metamorphosis, which produces an almost symmetrical adult animal (*Paris et al., 2008*).

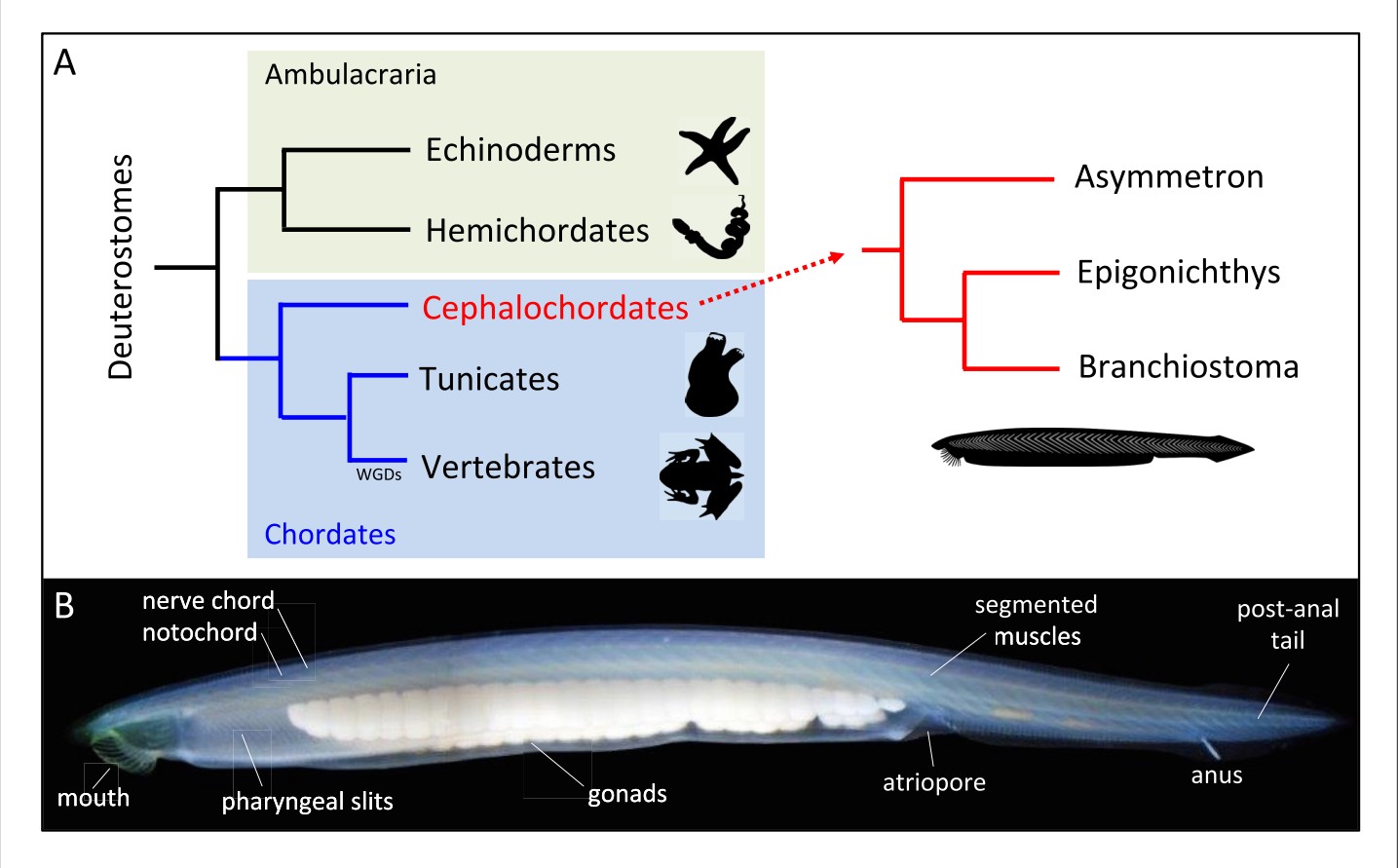

**Figure 1.** Deuterostome phylogeny and body plan for amphioxus. (**A**) Deuterostomes are subdivided into ambulacraria (echinoderms and hemichordates) and chordates (cephalochordates, tunicates and vertebrates). Cephalochordates, which are commonly known as amphioxus or lancelets, are further divided into three genera: *Branchiostoma*, *Epigonichtys* and *Asymmetron*. Whole genome duplication (WGD) occurred specifically in vertebrates. (**B**). Photograph of a *Branchiostoma lanceolatum* specimen exhibiting the typical body morphology shared by all cephalochordates. The body is elongated, with pointed extremities hence its name which comes from the Greek "amphi = both" and "oxus = pointed", and a series of chordate synapomorphies are indicated, such as the dorsal nerve chord and notochord, pharyngeal slits, segmented muscles and gonads, atriopore, caudal fin and post anal tail. Anterior is to the left and dorsal to the top.

The amphioxus genome also shows a high degree of conservation with vertebrate genomes, but with specific features. Amphioxus has orthologues for mostly all known vertebrate gene families, and the gene position and order in the genome, known as synteny, is also highly conserved which greatly benefits comparative analyses with vertebrates (*Putnam et al., 2008*; *Marlétaz et al., 2018*). However, the amphioxus genome has not undergone the two complete duplications that the vertebrate ancestor experienced (*Figure 1A*), although it has undergone numerous specific gene duplications (*Brasó-Vives et al., 2022*).

Moreover, the regulation of gene expression is much simpler than in vertebrates (*Gil-Gálvez et al., 2022*). And although it is still a point of debate, the amphioxus three-dimensional chromatin structure also seems to be less complicated than the vertebrate's one (*Acemel et al., 2016*; *Huang et al., 2023*). Altogether, the crucial phylogenetic position, conserved morphological traits and genome organization make amphioxus a useful organism for answering fundamental questions in biology, particularly with respect to vertebrate evolution. Thus, over the last decades, cephalochordates have become an important animal model in the fields of evolutionary developmental biology (EvoDevo), immune system evolution, cell signalling, regeneration and genome evolution.

## Systematics and diversity
The proposed phylogenetic position of cephalochordates has, as with many other metazoan groups, undergone major changes in recent years. Cephalochordates used to be classified

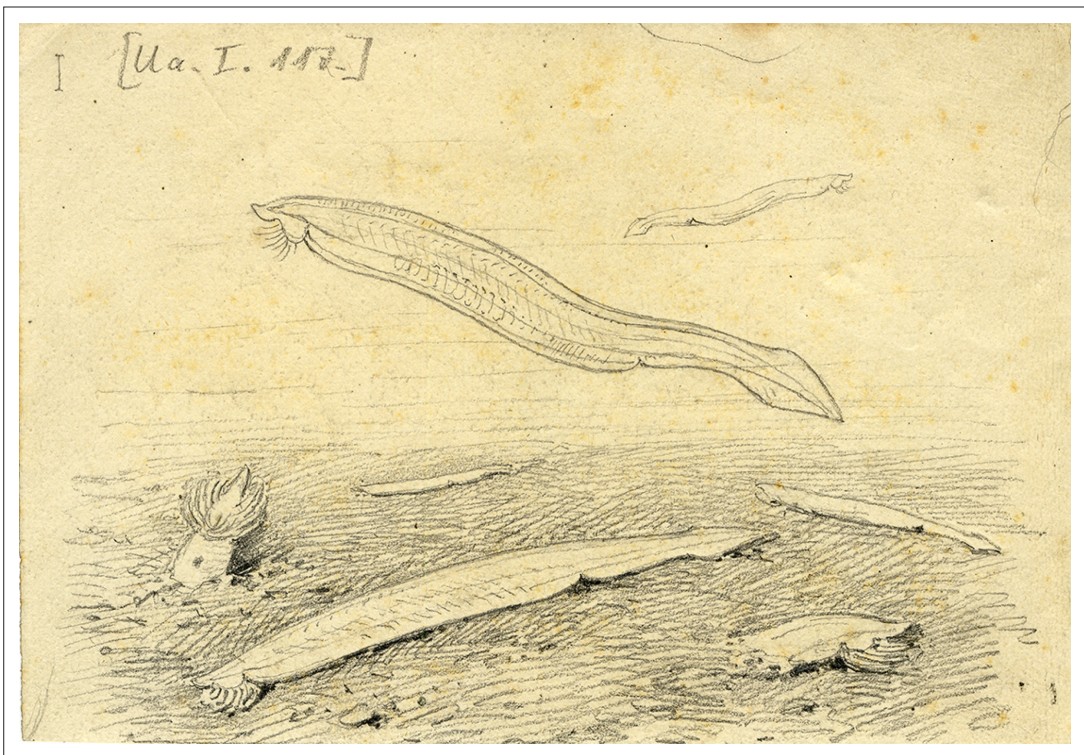

**Figure 2.** A drawing by Comingio Merculiano showing *Amphioxus lanceolatus* (now known as *Branchiostoma lanceolatum*). This drawing of adult amphioxus is based on research done at the Stazione Zoologica Anton Dohrn (SZN) between 1880 and 1890. Some of the amphioxus in the drawing are buried in the sand at the bottom of the sea, which is a relatively rare occurrence. The typical anatomical features of chordates (see *Figure 1B*) are clearly visible, which is a testament to the accuracy of Merculiano's drawings.

Image courtesy of Stazione Zoologica Anton Dohrn – Archivio Storico; used with permission (SZN 2022 (L6) Prot. n. 6630). This image is not covered by the CC-BY 4.0 license and further reproduction of this panel would need permission from the copyright holder.

as the closest group to vertebrates within the chordates, one of the two deuterostome clades together with the Ambulacraria (*Figure 1A*). This position, as a sister group of vertebrates, was based above all on the conservation of numerous morphological characteristics, and also on some molecular studies based on rDNA (*Winchell et al., 2002*). Moreover, the other chordate subphylum, the tunicates, shows a great divergence at the morphological level, especially in adults, whose body plan is completely different from that of the prototypical chordate.

However, this classification has completely changed following studies using larger molecular data, which finally positioned the cephalochordate lineage as the earliest divergent group of chordates (*Bourlat et al., 2006*; *Delsuc et al., 2006*; *Delsuc et al., 2008*), and placing the tunicates as the sister group of the vertebrates. This new chordate phylogeny suggests an evolutionary explanation of why tunicates, despite their tremendous anatomical and genomic divergence, share some features with vertebrates that

are absent in amphioxus such as migratory cells similar to those of the neural crest, or placode-like ectodermal regions (*Abitua et al., 2015*; *Manni et al., 2004*; *Horie et al., 2018*).

Another consequence of this newly proposed phylogenetic classification is a change in the hypothesis about the ancestral chordate life-style. In the past, it was hypothesised that vertebrates arose by neoteny from a sessile organism with free-living tadpole larvae (like ascidians) (*Williams, 1996*). Placing cephalochordates as the earliest divergent chordates suggests that instead the ancestral chordate could have been amphioxus-like with a free-living lifestyle even at the adult stage. This hypothesis is also reinforced by the fact that early vertebrate fossils, such as *Haikouichthys* or *Haikouella*, are similar to amphi-oxus in many aspects, such as their small size or their mobile and filter-feeding lifestyle (*Mallatt and Chen, 2003*; *Shu et al., 2003a*; *Shu et al., 2003b*).

Unfortunately, clear cephalochordate fossils have not yet been found. For many years,

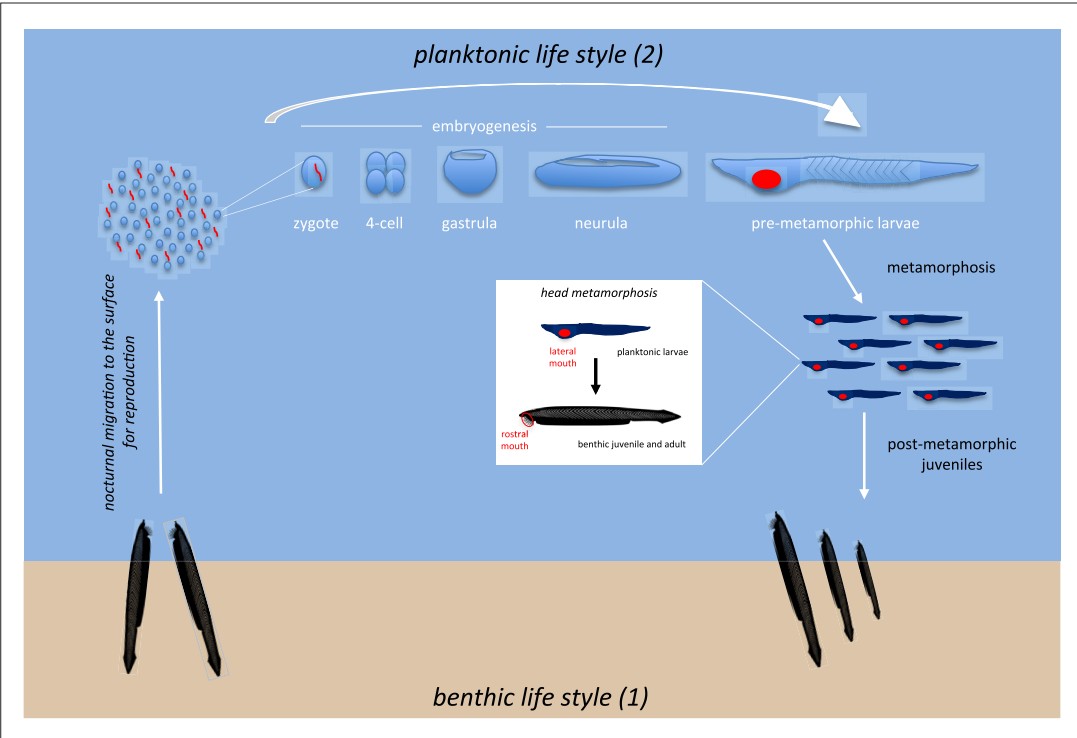

**Figure 3.** The life cycle of amphioxus includes benthic and pelagic phases. Adults live at the bottom of the sea during the benthic phase (1); external fertilization takes place in the water column during the breeding season; embryos and larvae then join the planktonic community near the surface during the pelagic phase (2). The length of time spent near the surface depends on the species (and varies from a few weeks to several months). Metamorphosis results in the mouth migrating from the left side of the larvae to a ventral and midline position (juvenile/adults). This is accompanied by a significant change in food intake strategy and a shift from pelagic to benthic behaviour.

*Pikaia*, the Cambrian fossil from Burgess Shale in Canada, has been considered a basal chordate, but numerous features such as dorsal organ, posterior ventral area, posterior fusiform structure, anterior dorsal unit, and sigmoid rather than chevron-shaped muscles, divide the paleontological community about the exact phylogenetic position of this controversial fossil, even if its unresolved position is most probably within chordates (**Morris and Caron, 2012**; **Mallatt and Holland, 2013**).

In 1774, the first scientific description of a specimen of amphioxus was made by Pallas from an animal from off the coast of Cornwall (United Kingdom) (**Pallas et al., 1774**). Years later, amphioxus were rediscovered in the Mediterranean Sea first by **Costa, 1834**, and, two years later, independently, by **Yarrell, 1836**. Apart from the difference in the nomenclature, the Cornish and Mediterranean specimens were considered the same species, *Branchiostoma lanceolatum*.

In 1847, Gray described a new specimen from the coast of Borneo, which he called *Branchiostoma belcheri* (**Gray, 1847**). Shortly

afterwards, in 1852, a new species was described off Peru by Sundevall and named *Branchiostoma elongatum* (**Sundevall, 1852**). The same year, *B. lanceolatum* was also observed off northern Germany (**Sundevall, 1852**). It was then that a system capable of species classification became necessary, and the enumeration of myotomes anterior to the atriopore, between the atriopore and anus, and posterior to the anus was chosen, the global morphology being extremely similar between all described cephalochordates. Thus, the classification soon included four species (*B. lanceolatum*, *B. belcheri*, *B. elongatum* and *Branchiostoma caribbaeum*, described from the coasts of Rio de Janeiro in Brazil, **Sundevall, 1853**), distributed in a rather cosmopolitan way between the Mediterranean, the Atlantic and the Pacific oceans.

In 1876, based on differences in pharyngeal slits position and fin shape in animals from Torres Strait in Australia, Peters described a second cephalochordate genus, which he called *Epigonichthys* (**Peters, 1876**). Although a later study reclassified the same species as belonging to the

genus *Branchiostoma* (*Günther and Reptilia, 1882*), *Epigonichthys* was later recognised as a new genus.

Finally, in 1893, Andrews described a third genus of cephalochordates, *Asymmetron*, for specimens from the Bahamas which he named *Asymmetron lucayanum* (*Andrews, 1893*). The main characteristics of this third genus were the presence of a single row of gonads and the asymmetrical metapleural folds, the left one ending at the level of the anus, while the right one was continuous behind with the ventral median fin. From this period on, many researchers began to describe new specimens from different locations and to define the classification of cephalochordates on the basis of the aforementioned meristic characteristics. This gave rise to discussions, with numerous synonyms for the same species or different species sharing the same name. Thus, new genera and subgenera were proposed, such as *Amphioxus*, *Heteropleuron*, *Amphioxides*, *Dolichorhynchus*, *Paramphioxus*, etc.

During this period, more than fifty species and ten genera of lancelets were described. Nevertheless, it was not until 1996 that Poss and Boschung made a compilation of all the described species and genera, and re-examined the different meristic data of each, to produce a list as correct as possible of the lancelet species described in the world (*Poss and Boschung, 1996*). This study reduced the total number of species to 29, and the number of genera to two, *Branchiostoma* and *Epigonichthys*. However, they defined *Asymmetron* as a synonym of *Epigonichthys*, since they failed to detect synapomorphies of the *Epigonichthys* group that would exclude *Epigonichthys lucayanum* (today called *Asymmetron lucayanum*) from this group. Thus, in the absence of arguments to support the fact that *E. lucayanum* (i.e. *A. lucayanum*) is the sister-taxon to all other *Epigonichthys*, they followed the classification proposed by *Richardson and McKenzie, 1994* with only two genera (*Epigonichthys* and *Branchiostoma*) instead of three genera as proposed in *Piyakarnchana and Vajropala, 1961*.

Poss and Boschung described the challenge of classifying the different species of amphioxus through the use of meristic data as follows: "Multivariate analysis of meristic variation, using primarily American species, reveals considerable intraspecific variability in key taxonomic features. Some species exhibit wide variation in countable segments, whereas others are characterized by a narrow range" (*Poss and Boschung, 1996*). Furthermore, they clearly advocated the use of molecular techniques capable of distinguishing

genetic differences and discriminating taxa with small morphological differences. In fact, it was through the use of modern molecular approaches, as well as through detailed morphological descriptions, that Nishikawa and Nohara confirmed the existence of three genera, *Branchiostoma*, *Asymmetron* and *Epigonichthys* (*Nishikawa, 2004*; *Nohara et al., 2005*). Two of them, *Asymmetron* and *Epigonichthys,* possess asymmetrical dextral gonads, and the third one, *Branchiostoma*, symmetrical gonads (*Igawa et al., 2017*).

The use of molecular taxonomy to define species has brought some other surprises in the classification of amphioxus species, since it revealed the existence of cryptic species among animals that previously shared the same name. Thus, for example, specimens of *B. belcheri* from the Chinese and Japanese coasts were differentiated into two different species (*B. belcheri* and *Branchiostoma tsingtauense*) (*Wang, 2004*; *Xu et al., 2005*) even if later, according to the rule of priority, the name of *B. tsingtauense* was changed by *B. japonicum* (*Wang, 2004*; *Zhang et al., 2006*).

Another example of a cryptic species complex, revealed through the use of molecular approaches, concerns *A. lucayanum,* for which up to three genetically distinguished major groups of geographical populations have been discovered. For one of these groups, composed of animals collected in the Red Sea, the name *Asymmetron rubrum* has even been proposed (*Subirana et al., 2020*; *Kon et al., 2006*). Thus, all these studies suggest that the total number of amphioxus species in the world is probably underestimated, and that molecular characterisation is likely to increase the total number of extant species in the near future.

## Distribution

The different amphioxus species can be found in a cosmopolitan way in all tropical and temperate oceans of the world. Amphioxus have never been observed in freshwater, and although they are present worldwide, they have a preference for soils of more or less fine sand or shell deposits, and in most cases with little organic decay (see below). However, they are not always present in all suitable sandy sediments, which indicate that other factors, such as pollution or currents, may play an important role in the dynamic distribution of these benthic animals. Of course, this distribution refers to adult animals, which have a benthic lifestyle and are generally found at depths

between very shallow water (i.e. 0.5 m deep) to 30–50 meters deep. There are also some exceptions since some specimens have been found at greater depths (i.e. about 180 m deep) (*Wickstead, 1975*) and in an anaerobic and sulfide-rich environment caused by the decomposing body of a whale at 229 m deep (*Nishikawa, 2004*), which does not exclude the possibility that deep-water species may be found in the future.

In contrast to adults, the embryonic and larval stages of amphioxus are planktonic. Thus, the amphioxus larvae can drift across oceans thanks to marine currents over a period of time that, depending on the species, can range from a couple of weeks to several months (*Figure 3*). The larvae are mainly found and distributed by coastal currents, although pelagic larvae have also been reported in places far away from the coast (*Goldschmidt, 1905*). The study of these larvae gave rise to the discussion of a different type of amphioxus, called Amphioxides, which were considered to be pelagic adults, although today there is a wide consensus on the larval nature of these individuals, which, however, have delayed their metamorphosis in a neotenic process (*Bone, 1957*).

The distribution of each amphioxus species has been reviewed in *Poss and Boschung, 1996*. This kind of information can also be accessed in the Unesco database, OBIS (*McEwan, 2001*), where it can be observed that amphioxus are found in all tropical and temperate coasts of the world. An interesting aspect of the differences in distribution between species is that, while some species have been found at very distant locations around the globe, such as *B. belcheri* that can be found in practically all coasts of East Asia, Oceania and even the African coasts of the Indian Ocean, and similarly *B. lanceolatum* that is found along the entire Mediterranean coasts, the Atlantic coasts of Europe and North Africa (*Caccavale et al., 2021b*), and even in the Indian Ocean, other species are found in much more restricted areas. For example, *Branchiostoma senegalensis* or *Branchiostoma gambiense* were only described on the West Africa coasts. Several explanations can account for this, but the most widely accepted is that the amplitude of the species distribution depend upon the type of marine currents present in each area (*Webb, 1975*).

## Habitat and lifestyle

Adult amphioxus, as already mentioned, live on the seafloor, burrowed in well-ventilated substrates with a soft texture and without too much organic load. Different species have been described as living in different types of substrate, ranging from very fine sand, coarse sand and even shell deposits, with a clear preference of most of the species for coarse sand with low content of fine particles. This is the case of *Branchiostoma nigeriense* on the west coast of Africa (*Webb and Hill, 1958*; *Webb, 1958*), *Branchiostoma caribaeum* in Mississippi Sound and from South Carolina to Georgia (*Boschung and Gunter, 1962*; *Cory and Pierce, 1967*), *B. senegalense* in the off-shore shelf region off North West Africa (*Gosselck and Spittler, 1979*) and *B. lanceolatum* from the Mediterranean coast of southern France (*Caccavale et al., 2021b*; *Desdevises et al., 2011*). However, *B. floridae* from Tampa Bay in Florida seems to be an exception to this rule since they live in fine sand bottoms (*Stokes and Holland, 1996a*; *Stokes, 1996*).

All species of amphioxus are gonochoric, and only a few cases of hermaphroditism have been reported in both *B. lanceolatum* and *B. belcheri* (*Yamaguchi and Henmi, 2003*; *Orton, 1914*). In these cases, only a few female gonads (i.e., developing ovaries) were observed in a male (2–5 gonads out of a total of 45–50). A unique case of complete sex reversal has been described in *B. belcheri*, where a female amphioxus reared in the laboratory was sexually reversed into a male (*Zhang et al., 2001*).

Spawning, which consists in the release of thousands of oocytes and millions of spermatozoa in the water column, is concentrated, in most species, in one period of the year (i.e. the spawning season), which usually takes place during the warmer months (spring-summer). The spawning season duration varies between species, being shorter (between one and three months) in species living in temperate waters and longer (around six months) in tropical species. Spawning always occurs shortly after sunset, although the behaviour is different depending on the species. Thus, for example, in *B. floridae*, up to 90% of the animals spawn synchronously once every two or three weeks (*Stokes and Holland, 1996b*). On the other hand, in other species such as *B. lanceolatum*, spawning occurs gradually between the beginning and the end of the spawning season (*Fuentes et al., 2004*). An exception to this unique annual breeding season is the case of *A. lucayanum*, which spawn during two periods of the year (during the warm months of spring and summer, but also in autumn). Moreover, in this species, the moon cycle seems to play a major role since spawning is concentrated

in the days preceding the new moon (*Holland, 2011*).

Concerning the feeding behaviour, the size of the particles filtered by different amphioxus species, as well as their diet, has been studied. Amphioxus are able to ingest sub-micron particles thanks to the mucus secreted by the endostyle. The size of these particles has been calculated in several species (i.e. *B. lanceolatum, B. senegalense, B. floridae*) and the results are quite similar regardless of the species. The size ranges from 0.062 to 100 μm, although in *B. senegalense* particles up to 300 μm were found (*Gosselck et al., 1978*; *Ruppert et al., 2000*; *Riisgard and Svane, 1999*). This particle size suggests that the amphioxus diet includes microbes as well as phytoplankton, even if, in addition to phytoplankton, crustaceans have also been found in the gut contents of *B. senegalense* and *B. lanceolatum* larvae (*Gosselck and Kuehner, 1973*; *Webb, 1969*). Moreover, much of the ingested material exits the anus undigested after 1–2 hours and most of the gut contents consist of detritus, suggesting that amphioxus are indiscriminate suspension feeders (*Gosselck et al., 1978*). A clear example of this indiscriminate filtering behaviour is the fact that several recent studies show how different species of amphioxus are capable of filtering microplastics present in the environment (*Cheng et al., 2023*; *Xiang et al., 2022*).

An interesting behaviour of adult amphioxus is that, as ciliary feeding progresses, the oral cirri, whose function is to prevent the entry of large particles, become blocked with these coarse detritus reducing the flow of water through the pharynx. When this occurs, the atrial floor is violently raised and lowered, and water is expelled from the atrium through the pharynx and oral hood, which unblocks the oral cirri (*Dennell, 1950*).

Several studies have focused on the lifespan of different amphioxus species, usually based on the size distribution of sampled individuals and taking into account that amphioxus grow continuously during their entire life (*Stokes, 1996*). These estimates include a lifespan of 2–3 years for *B. floridae* (*Wells, 1926*; *Nelson, 1968*; *Futch and Dwinell, 1977*), a maximum age of 2–3 years for *B. belcheri* (*Chin, 1941*; *Chen et al., 2008*), a lifespan of 4–5 years for *B. senegalense* in northwest Africa (*Gosselck and Spittler, 1979*), a lifespan of 5 years for *B. lanceolatum* in the Mediterranean Sea (*Desdevises et al., 2011*), which increases to 8 years in the relatively cold waters of Helgoland (*Courtney, 1975*).

The most likely causes of death in amphioxus, as in most wild animals, can be summarized as infections and predation. Thus, our own observations attest that amphioxus in the water column are attractive prey for fishes, and a description of amphioxus predators has been published. In this case, a stingray was observed to have a gut filled almost exclusively with amphioxus (*B. floridae*) in Tampa Bay (*Stokes and Holland, 1992*).

Concerning infections, in 1936 Ravitch-Stcherbo described the presence of a bacteria which produces a red pigment by putrid decomposition of tissues, and which is capable of infecting and killing amphioxus (*B. lanceolatum*) in captivity, but the strain of this bacterium was not described (*Ravitch-Stcherbo, 1936*). More recently, Zou and collaborators described the presence of a lethal bacteria and characterised it as *Vibrio alginolyticus* in *B. belcheri* (*Zou et al., 2016*). Finally, other causes of death like tumours, such as a chromaffinoma (*Stolk, 1961*), or the presence of parasites have also been described in amphioxus. Thus, in 1968 Azariah described the presence of a trypanorhynchan larvae, a cestode known to parasitize fishes, in several individuals of *B. lanceolatum* off the coast of Madras in India (*Azariah, 1968*), and Holland and collaborators also described the presence of parasitic larvae of the tapeworm *Acanthobothrium brevissime* in *B. floridae* (*Holland et al., 2009*).

## Technical advances in amphioxus research

In recent years, the worldwide growing interest in amphioxus as model organisms for different research studies has led various groups to develop new technical approaches to breed the animals in captivity. Thus, different tools and protocols have been developed for amphioxus maintenance and reproduction that allow obtaining large amounts of live embryos in the laboratory. These amphioxus aquaculture systems have been developed for the four most studied species, with slight differences concerning the day/night cycle, sea water recirculation, species-specific temperature regimes, natural or artificial seawater, the presence or not of sand in the tanks, and so forth (*Fuentes et al., 2004*; *Fuentes et al., 2007*; *Holland and Yu, 2004*; *Yasui et al., 2007*; *Holland and Holland, 2010*; *Li et al., 2012*; *Li et al., 2013*; *Li et al., 2015*; *Theodosiou et al., 2011*; *Benito-Gutiérrez et al., 2013*; *Carvalho et al., 2017*; *Somorjai et al., 2008*).

Adult amphioxus with mature gonads can be artificially induced to spawn in the laboratory

during the breeding season under controlled conditions. This is a prerequisite for the *in vitro* fertilization of eggs and the achievement of synchronized embryo's cultures. Different methodologies have proven effective for successful spawning induction depending on the species: the first of these was an electric shock in *B. floridae* (*Holland and Holland, 1989*), but this approach also induced unfertilised egg activation in other species, such as *B. lanceolatum*, so a different approach was required. A water temperature change 36 hours prior spawning is employed for species like *B. lanceolatum* (*Fuentes et al., 2004*; *Fuentes et al., 2007*). Gonad maturation is a prerequisite for spawning induction, but it is seasonally restricted to the breeding season and is often quite difficult to obtain in captive animals. Nevertheless, excellent results have been obtained using tropical species (*B. floridae* and *B. belcheri*) for which it has been possible to significantly increase the reproductive period artificially in the laboratory, beyond the limited breading season (*Li et al., 2013*; *Holland and Li, 2021*).

Animal husbandry, therefore, allows obtaining large amounts of eggs and embryos on demand, opening the door to modern functional approaches to study developmental gene function and the molecular mechanisms of gene and genomic regulation. The first studies focusing on gene expression using amphioxus embryos were based on classical analyses through in situ hybridization in the 1990s (*Holland et al., 1992*). The first functional studies were carried out through the use of pharmacological treatments capable of activating, inhibiting or modifying certain signalling pathways (*Bertrand et al., 2017*). Other methods to manipulate gene expression through gene overexpression or gene knockdown by microinjection in the unfertilized eggs of mRNAs or morpholinos have also been developed in different amphioxus species (*Aldea et al., 2019*; *Onai et al., 2010*; *Schubert et al., 2005*).

Classical embryo micromanipulation techniques, including grafting, have also been developed (*Le Petillon et al., 2020*). Importantly, through the use of the TALEN and CRISPR-Cas9 gene-editing approaches, and Tol2-based transgenesis, it has been possible to obtain knock out and transgenic lines in *B. floridae* and *B. belcheri* for different genes, which has lifted an important brake on functional studies using amphioxus and has boosted the research in the evolutionary developmental biology field (EvoDevo) (*Holland and Li, 2021*; *Li et al., 2014*; *Li et al., 2017*; *Hu et al., 2017*; *Zhong et al., 2020*; *Ren et al.,*

*2020*; *Zhu et al., 2020*; *Zou et al., 2021*; *Su et al., 2020*; *Kozmikova and Kozmik, 2015*).

Finally, high-throughput sequencing techniques have made it possible to obtain the complete chromosome-level genome assembly of four amphioxus species, *B. floridae, B. lanceolatum, B. belcheri* and *B. japonicum* (*Putnam et al., 2008*; *Marlétaz et al., 2018*; *Brasó-Vives et al., 2022*; *Huang et al., 2023*; *Huang et al., 2014*), thus opening the door to functional and comparative genomics studies. The use of new sequencing techniques at the level of single cells has been producing significant amount of information in recent years in various animal models. Amphioxus has not been left behind and this technique has also started to generate interesting results in several species (*Lin et al., 2020*; *Satoh et al., 2021*; *Ma et al., 2022*).

## Amphioxus as a model to understand chordate evolution

In this article we have focused mainly on known data on the biology and natural history of amphioxus. However, most of the recent scientific work published on amphioxus focuses on the evolution of developmental mechanisms and genomes. As we have presented in this review, because of their phylogenetic position among chordates, their prototypical characteristics, and the possibility of obtaining a large amount of externally developing and transparent embryos, amphioxus were mainly used to try to understand how the evolution of genomes and of the control of developmental processes led to the morphological complexity found in extant vertebrates.

Concerning genomics, obtaining whole genome sequences for amphioxus (*Putnam et al., 2008*; *Marlétaz et al., 2018*; *Huang et al., 2014*), and also for tunicates (*Dehal et al., 2002*; *Dehal and Boore, 2005*), allowed the 2R hypothesis proposed by *Ohno, 1970* to be confirmed. According to this hypothesis, two rounds of whole genome duplications took place during the early evolutionary history of vertebrates, although data in lamprey suggest that only one of these duplications might be shared by gnathostomes (jawed vertebrates) and cyclostomes (jawless vertebrates including lampreys and hagfish) (*Simakov et al., 2020*). Amphioxus genomic data also helped reconstructing the chordate ancestral karyotype, and the evolution of gene families in this clade. Finally, recent epigenomic analyses showed that chromatin conformation evolution (*Acemel et al., 2016*; *Huang et al., 2023*) and complexification of developmental gene

> ## Box 1. Outstanding questions about the natural history of amphioxus.
>
> # How much amphioxus diversity remains undiscovered? In other words, how many species are valid?
>
> # What are the ecological factors that restrict the distribution of amphioxus species to specific places?
>
> # What environmental, physiological and/or endocrine factors are responsible for the spawning induction in the wild?
>
> # Why do certain species develop gonads in captivity in a simple way, while this process is extremely complicated in other species?
>
> # What are the greatest threats to amphioxus conservation?
>
> # What mechanisms allow morphological and anatomical conservation between different amphioxus species despite their high genetic polymorphism?

regulation (*Marlétaz et al., 2018*) and of the interconnectivity between signalling pathways (*Gil-Gálvez et al., 2022*) might have participated to the emergence of vertebrate specific traits.

Studies of amphioxus development, through the analysis of gene expression or function, and of the role of different intercellular communication pathways, led to several key advances in our understanding of morphological evolution within the chordate group. First, conservation of the expression of orthologous genes in homologous structures between amphioxus and vertebrates allowed highlighting the key actors controlling the formation of chordate synapomorphic traits. Hence it has been shown, for example, that both amphioxus and vertebrates possess an embryonic territory at the gastrula stage called the dorsal organizer, which is responsible for early axial patterning and for neural induction (*Le Petillon et al., 2017*; *Yu et al., 2007*).

On the other hand, studies in amphioxus also pointed out differences with vertebrate developmental modalities that could be linked to the emergence of vertebrate traits such as an unsegmented head musculature (*Aldea et al., 2019*; *Bertrand et al., 2011*; *Meister et al., 2022*). Lastly, embryological studies on amphioxus may also shed light on unsuspected roles of certain signals in the control of chordate development. For example, a recent work showed the role of the nitric oxide pathway in normal pharyngeal development through an interaction with the retinoic acid signaling pathway in amphioxus (*Caccavale et al., 2021a*), which calls for a more detailed examination of embryonic function of nitric oxide in vertebrates.

## Conclusions

The interest in the study of cephalochordate biology and ecology has experienced alternating periods of great popularity and long periods of stagnation. Today, however, the number of research groups using amphioxus as a model organism is growing and they are located all over the world. Moreover, the interest in amphioxus covers a wide spectrum of research fields, ranging from classical embryology, through EvoDevo, to functional and comparative genomics (see *Box 1* for a list of outstanding questions about the natural history of amphioxus).

Unlike many other animal models, there is not a specific meeting for researchers interested in amphioxus, although the European Society for Evolutionary Developmental Biology (https://evodevo.eu) has sponsored a satellite meeting dedicated to amphioxus for the past decade. This meeting, which takes places every two years, is typically attended by almost one hundred participants.

Modern technological approaches are driving amphioxus research, and undoubtedly the technical developments we have discussed, and in particular the obtaining of the transcriptomic profile of each cell type, will help resolving old questions in the future. Examples of such questions include the evolutionary appearance of the neural crest cells typical of vertebrates, or the evolution and complexification of the vertebrate brain from that of the chordate ancestor. The multiplication of data produced by high-throughput sequencing techniques will also certainly raise new and interesting questions, as much on the evolutionary level as on

that of embryonic development or physiology of amphioxus.

Whatever the future of research using different species of amphioxus as model organisms, one point on which the entire scientific community agrees is that this is an extremely exciting time to be working with this fascinating little animal.

## Acknowledgements
The authors thank present and past members of the Stazione Zoologica Anton Dohrn (SZN) and Observatoire Océanologique, Banyuls-sur-Mer (OOB) for their enthusiasm and work on several amphioxus research projects. The authors also thank the reviewers for helping to improve the manuscript.

**Salvatore D'Aniello** is in Biology and Evolution of Marine Organisms (BEOM), Stazione Zoologica Anton Dohrn, Napoli, Italy
salvatore.daniello@szn.it
https://orcid.org/0000-0001-7294-1465
**Stephanie Bertrand** is at Sorbonne Université, CNRS, Biologie Intégrative des Organismes Marins (BIOM), Observatoire Océanologique, Banyuls-sur-Mer, France
https://orcid.org/0000-0002-0689-0126
**Hector Escriva** is at Sorbonne Université, CNRS, Biologie Intégrative des Organismes Marins (BIOM), Observatoire Océanologique, Banyuls-sur-Mer, France
hector.escriva@obs-banyuls.fr
https://orcid.org/0000-0001-7577-5028

*Author contributions:* Salvatore D'Aniello, Conceptualization, Data curation, Supervision, Writing – original draft, Writing – review and editing; Stephanie Bertrand, Conceptualization, Data curation, Writing – review and editing; Hector Escriva, Conceptualization, Data curation, Supervision, Writing – original draft, Writing – review and editing

*Competing interests:* The authors declare that no competing interests exist.

## Funding

| Funder | Grant reference number | Author |
|--------|------------------------|--------|
| Agence Nationale de la Recherche | ANR-19-CE13-0011 | Stephanie Bertrand Hector Escriva |
| Agence Nationale de la Recherche | ANR-21-CE13-0034 | Stephanie Bertrand Hector Escriva |
| Stazione Zoologica Anton Dohrn | | Salvatore D'Aniello |

| Funder | Grant reference number | Author |
|--------|------------------------|--------|

The funders had no role in study design, data collection and interpretation, or the decision to submit the work for publication.

## Decision letter and Author response
Decision letter https://doi.org/10.7554/eLife.87028.sa1
Author response https://doi.org/10.7554/eLife.87028.sa2

## Data availability
No new data was generated for this article.

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
