## [Decision Letter]

**Decision letter after peer review:**

Thank you for submitting your article "The cephalochordate amphioxus as a model to study vertebrate development and evolution" to *eLife* for consideration as a Feature Article. Your article has been reviewed by three peer reviewers, and the evaluation has been overseen by a member of the *eLife* Features Team (Peter Rodgers). The following individuals involved in review of your submission have agreed to reveal their identity: JK Sky Yu, Tokiharu Takahashi, and Filipe Castro.

The reviewers and editors have discussed the reviews and we have drafted this decision letter to help you prepare a revised submission.

Summary:

This manuscript provides a valuable review of the scientific history, basic biology, and recent research on cephalochordates, focusing on the contributions of amphioxus as a model for comparative genomics and evolutionary developmental biology. Recent technological advances have taken amphioxus research to the next level, and this review will undoubtedly be valuable to those seeking background knowledge on amphioxus. The key outstanding questions regarding amphioxus biology are also well organized and provide an excellent research framework for coming years. However, there are a number of points that need to be addressed to make the article suitable for publication.

Essential revisions:

1) Line 83-84; a recent paper (Huang et al. 2023 PNAS 120 e2201504120) has shown that the amphioxus Hox cluster is also divided into two TAD domains from gastrula stage, similar to that in vertebrates. Thus, the 3-D chromatin architecture of amphioxus genome may not be much simpler than that of vertebrates.

2) Line 98-100. Here the authors only cite Delsuc et al. 2006 Nature paper in this sentence; however, in this paper the phylogenetic analysis did not recover the monophyletic chordate clade and grouped cephalochordates with echinoderms. A later paper by Bourlat et al., (Nature 444:85-88, 2006) corrected this problem by including more data from invertebrate deuterostome animals for phylogenetic analyses, and further support the Olfactores (tunicates + vertebrates) hypothesis. Subsequently Dulsuc et al. also published another paper (genesis, 46:592-604, 2008) with broader sampling to correct their previous analysis, and they also recovered the monophyletic chordate group and further support cephalochordates as the most basal lineage while tunicates and vertebrates are sister groups. I would suggest the authors to also cite these later publications for completeness of information.

3) Line 102-104; I would suggest the authors to include some appropriate references here about those traits that are shared between tunicates and vertebrates but absent in cephalochordates. About MHB evolution, I think it is still a contentious issue. There are expression data suggesting a MHB-like region exists in amphioxus CNS (Castro 2006 Dev Biol 295:40-51; Albuixech-Crespo 2017 PLoS Biol 15: e2001573), but whether it has secondary organizer function remains unknown. In addition, a MHB-like gene regulatory program appears to be present in hemichordate S. kowalevskii (Pani 2012 Nature 483:289-2924), suggesting a conserved gene module for AP patterning of the embryonic ectoderm was already evolved in the deuterostome common ancestor.

4) Line 163-164; the last part of this sentence is incorrect. Both Asymmetron and Epigonichthys have gonads only on the right side of the animals, whereas Branchiostoma has one row of gonads on each side of the body (see references such as: Igawa 2017 Sci Rep 7: 1157; Holland and Holland 2022 Curr Topics Dev Bio 147:563-594).

5) Line 318-320; The paper in PNAS by Huang et al. mentioned above has further increased the number of chromosome-level genome assemblies to four amphioxus species, as well as Hi-C data from B. floridae, B. japonicum, and B. belcheri. The authors should include these new results into this review.

In addition, there are quite a few valuable bulk RNA-seq transcriptome datasets from several Branchiostoma and Asymmetron species, which have contributed extensively to many comparative analyses and phylogenomics studies. Again, the authors should consider mention this work.

6) Lines 286-323 ("Technical advances in amphioxus research")

In this section, the authors first present a chronological overview of amphioxus husbandry and spawning induction techniques. They then summarise the technical advances in functional experiments, most of which have been made in recent years. However, they overlook the descriptive studies, which, although they may appear old-fashioned, have undoubtedly contributed to and still play an important role in amphioxus research. In order to give the readers a context for understanding the whole picture of modern amphioxus studies, it would be helpful to include a short paragraph discussing the introduction of molecular biology (mainly in situ hybridisation) to amphioxus research, citing some representative seminal papers such as Holland PW et al. (Development 116:653-61), which started the amphioxus EvoDevo field.

7) In addition, for transgenic amphioxus lines (line 315), it is necessary to include the Tol2 assay (TALEN and CRISPR are for knock out). More detailed information on TALEN and Tol2 techniques can be found in Chapter1 "Laboratory Culture and Mutagenesis of Amphioxus" by Linda Holland and Guang Li in Developmental Biology of the Sea Urchin and Other Marine Invertebrates (pp1-9, Humana Press), so citing this reference may be useful. The same applies to Su L et al. Genes 2020, 11, 1311; doi:10.3390/genes11111311 for the CRISPR technique.

8) Lines 337-369 ("Amphioxus as a model to understand vertebrate evolution")

This section is shorter than expected and would benefit from a more in-depth description of recent results and their impact on our understanding of vertebrate evolution (including, possibly, one or two extra figures).

9) Please consider adding a figure which shows the global distribution of the species discussed in the article.

10) The reviewers also felt that the title needed attention and the editor will discuss a new title with you if the revised article is accepted for publication.

---

## [Author Response]

Essential revisions:1) Line 83-84; a recent paper (Huang et al. 2023 PNAS 120 e2201504120) has shown that the amphioxus Hox cluster is also divided into two TAD domains from gastrula stage, similar to that in vertebrates. Thus, the 3-D chromatin architecture of amphioxus genome may not be much simpler than that of vertebrates.

We appreciate this comment from the reviewers, however, the results of Huang et al., although interesting, as they open a discussion on the evolution of genomic structure in chordates, present some questionable points concerning the presence of two TADs in the Hox locus in amphioxus. It is one thing to say that there is a TAD boundary in the middle of the amphioxus Hox locus (very debatable on the other hand by looking at the HiC results of Huang et al.) and another very different is to say that the general 3D structure is conserved with vertebrates. The anterior region is syntenically equivalent to vertebrates (as we published in Acemel et al. NatGen). However, the "posterior" TAD of amphioxus (assuming that the TAD from Hox6-7 onwards is a boundary TAD) only includes up to Hox15 and EvxA-B (at most), which in vertebrates is attached to the cluster. The entire syntenic region to the vertebrate posterior TAD is outside in the amphioxus described TAD (i.e. particularly the introns of Jazf where the Hox enhancers are described in vertebrates). Lnp is also out and 80% of the region orthologous to the vertebrate TAD is on another chromosome in amphioxus. Thus, the results, that under our interpretation are weak, concerning the TAD boundary together with the lack of conservation of the composition of the posterior TAD (including the Hox regulatory regions) lead us to disagree with the results presented by Huang and colleagues.

However, since this article is not intended to discuss this point, we have changed the text to include the possible debate between the two studies, so that the reader can access both interpretations.

2) Line 98-100. Here the authors only cite Delsuc et al. 2006 Nature paper in this sentence; however, in this paper the phylogenetic analysis did not recover the monophyletic chordate clade and grouped cephalochordates with echinoderms. A later paper by Bourlat et al., (Nature 444:85-88, 2006) corrected this problem by including more data from invertebrate deuterostome animals for phylogenetic analyses, and further support the Olfactores (tunicates + vertebrates) hypothesis. Subsequently Dulsuc et al. also published another paper (genesis, 46:592-604, 2008) with broader sampling to correct their previous analysis, and they also recovered the monophyletic chordate group and further support cephalochordates as the most basal lineage while tunicates and vertebrates are sister groups. I would suggest the authors to also cite these later publications for completeness of information.

That is correct, thanks, we added the indicated references.

3) Line 102-104; I would suggest the authors to include some appropriate references here about those traits that are shared between tunicates and vertebrates but absent in cephalochordates. About MHB evolution, I think it is still a contentious issue. There are expression data suggesting a MHB-like region exists in amphioxus CNS (Castro 2006 Dev Biol 295:40-51; Albuixech-Crespo 2017 PLoS Biol 15: e2001573), but whether it has secondary organizer function remains unknown. In addition, a MHB-like gene regulatory program appears to be present in hemichordate S. kowalevskii (Pani 2012 Nature 483:289-2924), suggesting a conserved gene module for AP patterning of the embryonic ectoderm was already evolved in the deuterostome common ancestor.

We thank the reviewer for highlighting this point. However, as the idea of this article is not to discuss possible controversial points, but rather, particularly in this paragraph, to indicate that certain morphological characteristics are shared by tunicates and vertebrates but absent in amphioxus, we have preferred to substitute the existence, or not, of a MHB, by the occurrence of placodes, structures that clearly exist in tunicates but not in amphioxus.

4) Line 163-164; the last part of this sentence is incorrect. Both Asymmetron and Epigonichthys have gonads only on the right side of the animals, whereas Branchiostoma has one row of gonads on each side of the body (see references such as: Igawa 2017 Sci Rep 7: 1157; Holland and Holland 2022 Curr Topics Dev Bio 147:563-594).

We agree and modified the text, thanks.

5) Line 318-320; The paper in PNAS by Huang et al. mentioned above has further increased the number of chromosome-level genome assemblies to four amphioxus species, as well as Hi-C data from B. floridae, B. japonicum, and B. belcheri. The authors should include these new results into this review.In addition, there are quite a few valuable bulk RNA-seq transcriptome datasets from several Branchiostoma and Asymmetron species, which have contributed extensively to many comparative analyses and phylogenomics studies. Again, the authors should consider mention this work.

Right, we now included the chromosome-level genomes available for other species.

6) Lines 286-323 ("Technical advances in amphioxus research")In this section, the authors first present a chronological overview of amphioxus husbandry and spawning induction techniques. They then summarise the technical advances in functional experiments, most of which have been made in recent years. However, they overlook the descriptive studies, which, although they may appear old-fashioned, have undoubtedly contributed to and still play an important role in amphioxus research. In order to give the readers a context for understanding the whole picture of modern amphioxus studies, it would be helpful to include a short paragraph discussing the introduction of molecular biology (mainly in situ hybridisation) to amphioxus research, citing some representative seminal papers such as Holland PW et al. (Development 116:653-61), which started the amphioxus EvoDevo field.

Thanks. We now added a sentence introducing the importance of molecular biology (mainly *in situ* hybridization) for the amphioxus research in the 90s.

7) In addition, for transgenic amphioxus lines (line 315), it is necessary to include the Tol2 assay (TALEN and CRISPR are for knock out). More detailed information on TALEN and Tol2 techniques can be found in Chapter1 "Laboratory Culture and Mutagenesis of Amphioxus" by Linda Holland and Guang Li in Developmental Biology of the Sea Urchin and Other Marine Invertebrates (pp1-9, Humana Press), so citing this reference may be useful. The same applies to Su L et al. Genes 2020, 11, 1311; doi:10.3390/genes11111311 for the CRISPR technique.

We thank the reviewer and added the two suggested references.

8) Lines 337-369 ("Amphioxus as a model to understand vertebrate evolution")This section is shorter than expected and would benefit from a more in-depth description of recent results and their impact on our understanding of vertebrate evolution (including, possibly, one or two extra figures).

We believe that the purpose of the *eLife* section “Natural History of Model Organism” is mainly to describe the general aspects of the biology of a given animal model. So in agreement with the *eLife* Editors when they accepted our review proposal, we decided that the important points of the article would focus on the known data on the ecology, life form, distribution and number of known species of amphioxus. The last point, about the usefulness of amphioxus as a model of study in EvoDevo, is simply to give an idea to the reader who does not know this animal of why it is important for the research without going into details, which, moreover, have been discussed in other reviews.

9) Please consider adding a figure which shows the global distribution of the species discussed in the article.

We thank the reviewer, nevertheless we prefer not to include this suggestion because such a figure is already published in Poss and Boschung (1996), that we have referenced in our work.

10) The reviewers also felt that the title needed attention and the editor will discuss a new title with you if the revised article is accepted for publication.

We agree and propose the following new title:

The Natural History of Model Organisms: The amphioxus as a model to study evolution of development in chordates